# Synergistic Effects of Multiple Heterojunctions and Dopant Atom for Enhancing the Photocatalytic Activity of C-Modified Zn-Doped TiO$_2$ Nanofiber Film

Ying Lu [1,2], Xiangge Qin [1,*] and Jinzhong Hong [3]

1 School of Materials Science and Engineering, Jiamusi University, Jiamusi 154007, China
2 School of Science, Jiamusi University, Jiamusi 154007, China
3 School of Civil Engineering and Architecture, Jiamusi University, Jiamusi 154007, China
* Correspondence: qinxiangge@jmsu.edu.cn

**Abstract:** To design efficient photocatalytic systems, it is necessary to inhibit the compounding of electron-hole pairs and promote light absorption in photocatalysts. In this paper, semiconductor heterojunction systems of C-modified Zn-doped TiO$_2$ composite nanomaterials with nanofiber structures were synthesized by electrospinning and hydrothermal methods. The composite nanofiber film was thoroughly characterized and the morphology, structure, chemical phases and optical properties were determined. Scanning electron microscopy confirmed that the nanofiber diameter was 150–200 nm and the C particles were uniformly modified on the smooth nanofiber surfaces. X–ray diffraction patterns and Raman show TiO$_2$ as a typical anatase, modified C as graphite and Zn as ZnOcrystals. Moreover, the entry of Zn and C into the TiO$_2$ lattice increases the crystal defects. Meanwhile, TiO$_2$, ZnO and graphite form multiple heterojunctions, providing pathways for photogenerated carrier transfer. These synergistic effects inhibit the recombination of electron-hole pairs and provide more reaction sites, thus improving the photocatalytic efficiency. UV-Vis diffuse reflectance spectroscopy and fluorescence spectroscopyimply that these synergistic effects lead to improved optical properties of the composite. Using organic dyes (methylene blue, methyl orange, rhodamine Bandmalachite green) as simulated pollutants, the composite nanofiber film exhibited good photocatalytic activity for all dyes due to the significantly large specific surface area, small size effect and synergistic effects of multiple heterojunctions and dopant atom. In addition, the nanofiber film has good reusability and stability for the photodegradation of organic dyes, so it has potential for industrial applications.

**Keywords:** nanofiber film; photocatalysis; heterojunction; dopant

## 1. Introduction

Given the rising concern about environmental pollution, there is an urgent need for efficient and sustainable technologies to reduce the pollutant content in wastewater [1–4]. Although organic dyes are not widely used, they are present mainly in the industrial wastewater of plastics, chemicals and dyeing, etc. Residual dyes in wastewater, although at low concentrations, are harmful to animals, plants and humans due to their non-degradability and toxicity [5–7]. Because of their unique physicochemical characteristics, nanoscale semiconductor photocatalytic materials are widely used in dye removal [8]. In recent decades, TiO$_2$ has become an attractive semiconductor for photocatalysis due to its favorable stability and environmental friendliness [9–12]. Unfortunately, TiO$_2$ almost always suffers from high recombination and low mobility of photogenerated charge carriers, low light absorption and poor specific surface area, resulting in reduced photocatalytic activity [13–15]. Heterojunctions provide new ideas for coupling two or three different semiconductors together to design composite photocatalysts. Among the strategies to

promote space charge separation and inhibit charge complexation, the heterojunction strategy proved to be effective because the internal electric field provides the driving force at the heterojunction region [16,17]. Under photoexcitation, the electric field force at the heterojunction interface causes holes to begin moving in the negative side and electrons to begin moving in the positive side, leading to the spatial separation of photogenerated electron-hole pairs of carriers, consequently enhancing their photocatalytic activity [18–20]. The graphite crystal formed by C is a zero energy level semiconductor material. When it forms a heterojunction with other semiconductor materials, photogenerated electrons can be transported and transferred to the semiconductor surface using C as a potential ladder, preventing the recombination of photogenerated electron-hole pairs. In addition, C nan sheets, C nanotubes, C nanoparticles and C nanowires all have graphitic structures containing many sp2-hybridized C atoms, which favorably delay the recombination time of photogenerated electron-hole pairs for photocatalytic efficiency enhancement [21]. Many studies have proved that the recombination of $TiO_2$ with C effectively enhances the photoactivity, commonly due to excellent electrical conductivity and additional light absorption of C [22]. Meanwhile, $ZnO/TiO_2$ heterojunctions with high photocatalytic activity were designed by different methods, mainly because of their easy separation of carriers [23–25]. Therefore, some studies have reported combining $ZnO/TiO_2$ and $C/TiO_2$ as composite photocatalytic materials [26,27], and the structural design of photocatalysts is an important method to improve photocatalytic activity. Compared to bulk materials, nanomaterials can increase the number of exposed surfaces, which are key to the adsorption and activation of reactant molecules and act as catalytically active sites for redox reactions. Moreover, the small size effect and quantum size effect of nanomaterials are obvious, which can improve the semiconductor energy band structure [28–31]. However, nanopowders have difficulties in recycling and tend to form agglomeration in water, affecting photocatalytic activity. Nanofiber film has the properties of nanomaterials while being easily separated from pollutants by short precipitation. Thus, there is a need to explore suitable nanostructured photocatalysts. Electrospinning has proven to be a reliable and simple method for the preparation of nanofibers [32]. Due to the advantages of their large specific surface area, low aggregation, easy recycling and significant small-size effect, electrospun nanofibers are widely used for the coupling of various functional materials [33–36]. Therefore, C-modified Zn-doped nanofiber materials were designed with the advantages of nanomaterials. This novel structure of the material with small particles of C is uniformly modified on the surface of the nanofibers without completely covering the surface of $TiO_2$. The heterojunctions formed by these small particles of C promote the $TiO_2$ photocatalytic reaction without reducing the contact area of $TiO_2$ with pollutants too much, which is beneficial to the photocatalytic reaction.

## 2. Materials and Methods

### 2.1. Materials

All chemicals used were commercially available and used directly without any further purification. Tetrabutyl titanate (99.7%, CAS 5593-70-4), polyvinylpyrrolidone (PVP) ($M_w$ = 1-300-000, CAS 9003-39-8) and zinc acetate dihydrate (98.0%, CAS 5970-45-6) were purchased from Sigma-Aldrich (St. Louis, MO, USA). Methylene blue (90.0%, CAS 7220-79-3), methyl orange (98.0%, CAS 547-58-0), rhodamine B (99.0%, CAS 81-88-9) and malachite green (AR, CAS 2437-29-8) were purchased from Shanghai Macklin Biochemical Technology Co., Ltd. (Shanghai, China). Absolute ethanol (99.5%, CAS 64-17-5), glacial acetic acid (99.5%, CAS 64-19-7) and N,N-Dimethylformamide (DMF) (99.5%, CAS 68-12-2) were purchased from Tianjin Kaitong Chemical Reagent Co., Ltd. (Tianjin, China). Deionized water was used for all experiments.

### 2.2. Synthesis of Zn-Doped $TiO_2$ Nanofiber Film

To prevent the hydrolysate of tetrabutyl titanate from clogging the nozzle during the electrospinning process, the nozzle was modified, as shown in Figure 1. The plastic conical

nozzle had a larger diameter than the needle to improve clogging, while the soft metal wire replaced the metal needle as the electrode. Compared to metal needles, which could easily react chemically with acid, plastic nozzles were resistant to acid corrosion, so they didnot contaminate the electrospinning solution. Conical shape facilitated the flow of the solution in the nozzle. To improve efficiency, multi-electrospinning nozzles were used for electrospinning.

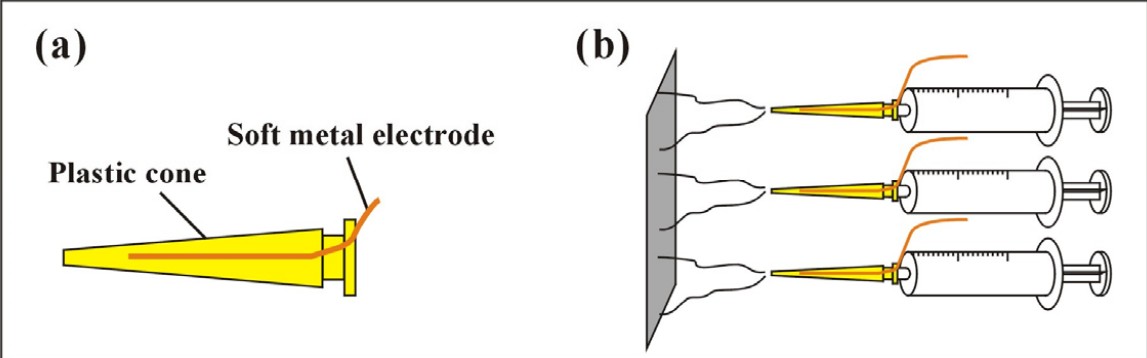

**Figure 1.** Basic set-up for (**a**) anti-clogging nozzle; (**b**) multi-electrospinning nozzles.

To remove water from crystallization, zinc acetate dihydrate ($Zn(CH_3COO)_2\cdot 2H_2O$) was dried at 100 °C for 1 h. Tetrabutyl titanate ($Ti(OC_4H_9)_4$) was used as a $TiO_2$ precursor, and ethanol absolute ($C_2H_6O$) was used as a solvent. Glacial acetic acid had to be used as part of the solvent to inhibit the hydrolysis of the tetrabutyl titanate. A solvent consisting of 4.0 mL of glacial acetic acid and 40.0 mL of absolute ethanol was added to a flask with a lid. Then, 4.0 mL of tetrabutyl titanate and 0.16 g of zinc acetate ($Zn(CH_3COO)_2$) were dissolved in the solvent and subsequently stirred for 10 min until it became a homogeneous solution. To avoid agglomeration, 3.2 g of polyvinyl pyrrolidone powder (PVP) was slowly poured into the above solution with stirring. Then, the mixture was stirred for 5 h to dissolve all the PVP to form a homogeneous and stable pale yellow solution at room temperature. For comparison, the reference sample (pure $TiO_2$) was synthesized by the same method.

The above 2 mL solution was placed in an 8 mL syringe with a plastic nozzle for electrospinning. The receiver of the nanofiber film was a piece of aluminum foil taped to a stainless steel plate, which was connected to the negative terminal of a high-voltage power supply. The electrostatic spinning voltage was set to a positive voltage +10 kV and a negative voltage −10 kV. The distance was about 10 cm from the aluminum foil to the nozzle tip. The nanofibers formed a white film containing PVP on the aluminum foil. The electrospun Zn-doped $TiO_2$ and pure $TiO_2$ nanofiber films were calcined in the air. The rate of calcination temperature rise was 25 °C/h until the heating was stopped at 500 °C, then held for 1 h to remove PVP, followed by cooling naturally to room temperature. The large film was broken into smaller films by calcination. Pure $TiO_2$ and Zn-doped$TiO_2$nanofiber films were denoted as T0 and TZ, respectively.

### 2.3. Loading of C onto Zn-Doped TiO_2 Photocatalyst

C was loaded onto Zn-doped$TiO_2$nanofiber film using hydrothermal and calcined methods. Next, 0.1 g of $TiO_2$ or Zn-doped $TiO_2$ nanofiber film was dispersed into 30 mL of absolute ethanol and deionized water (1:2 *v/v*) solution. Then, 10 mL DMF was added to the above system together with 2 mL of acetic acid. The obtained mixture was placed in a stainless steel hydrothermal reactor, which had a Teflon-lined vessel with a lid. T0 and TZ samples were hydrothermally treated at 180 °C for 24 h. To remove excess interferences in the hydrothermal solution, the precipitates were washed 6 times with deionized water after the hydrothermal treatment. Finally, the obtained precipitates were calcined in nitrogen at 400 °C for 1 h. C was based on $TiO_2$ and Zn-doped $TiO_2$ nanofiber films denoted as TC and

TZC, respectively. Figure 2 further illustrates the preparation method and process of TZC composite nanofiber film.

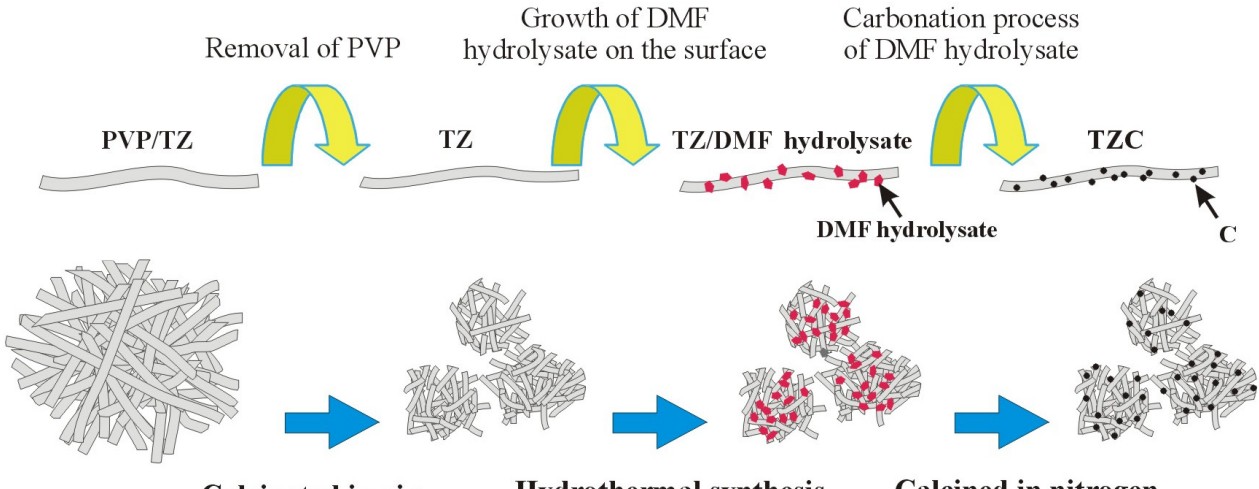

**Figure 2.** Preparation process of composite nanofiber film materials.

*2.4. Characterization*

All sample characterization methods, equipment, product models and countries of production are shown in Table 1.

**Table 1.** Information of characterization methods, equipment, product models and countries.

| Characterization Methods | Equipment | Product Model | Country |
| --- | --- | --- | --- |
| Scanning electron microscopy (SEM) Energy-dispersive X-ray spectrometer (EDS) | Field-emission scanning electron microscope | Jeol JSM-7800F | Japan |
| Transmission electron microscopy (TEM) High-resolution transmission electron microscopy (HR-TEM) | Field-emission transmission electron microscope | Jeol JEM-2100F | Japan |
| X–ray diffraction (XRD) patterns | X-ray powder diffractometer | Bruker D8 Advance | Germany |
| Raman spectra | Microscopic confocal Raman spectrometer | Renishaw inVia | UK |
| X–ray photoelectron spectroscopy (XPS) | X-ray photoelectron spectrometer | ThermoEscalab 250 Xi | USA |
| UV-Vis diffuse reflectance spectroscopy (UV-Vis DRS) | UV-Vis spectrophotometer | Shimadzu UV-3600 | Japan |
| Fluorescence spectra (FS) | Fluorescence spectrophotometer | Gangdong F-380 | China |

The morphological characteristics of the composite nanofiber film were elucidated by SEM analysis equipped with EDS. TEM and HR-TEM observations were carried out at an acceleration voltage of 200 kV.

The structure and crystal phase were analyzed by XRD with Cu Kα radiation. The X-ray tube operated at a voltage of 40 kV and produced a wavelength of 0.1542 nm. XRD scanned all samples $2\theta$ in the angular range of 20–80°, while the angular speed was 6°/min. The Raman spectra were conducted with an excitation of 325 nm laser light.

Chemical states of elements were accomplished by XPS. Al-K$\alpha$ irradiation was used in the test.

Optical absorption properties were detected by UV-Vis DRS with $BaSO_4$ as a reference in the scanning range of 200–800 nm. FS were detected with a maximum excitation wavelength of 300 nm. The scanning wavelength range was 330–520 nm.

*2.5. Photocatalytic Activity*

Photodegradation experiments were conducted to evaluate the photocatalytic activity of the prepared nanofiber films. The four organic dyes used as simulated pollutants for the experiments were methylene blue (MB), methyl orange (MO), rhodamine B (RhB) and malachite green (MG). First, 50 mg of nanofiber film (T0, TZ, TC or TZC) was dispersed into 50 mL of MB, MO, RhB or MG solution (10 mg/L). A catalyst content of 1 g/L in the dye was the best compromise for an optimal kinetic study. To eliminate the effect of adsorption, each nanofiber film was stirred continuously in the dark for 10 min to maintain adsorption–desorption equilibrium. The temperature was maintained at $30 \pm 2$ °C using a thermostatic bath. The pH was the natural pH of the dye solutions. The mixtures were irradiated with a 350 W Xe lamp used as the simulated sunlight. At intervals of 10 min or 5 min, 3 mL of organic dye was collected and centrifuged to separate the solid nanofiber film from the solution quickly. The clear solutions were analyzed by a spectrophotometer to measure the residual concentration. After the photocatalytic experiment, the photocatalyst was recovered by precipitation. The centrifuged and precipitated photocatalysts were combined for the next cycle of experiments after being washed and dried. The same volume of organic dye was used for each cycling experiment. The reusability of C-modified Zn-doped $TiO_2$ was examined by five consecutive cycles in photocatalytic degradation.

## 3. Results and Discussion

To observe the lattice and the morphology of nanofiber films, TEM and SEM with the corresponding EDS spectra were used. To increase the electrical conductivity, the samples for SEM were coated with Pt. As shown in Figure 3a,b, the prepared PVP/TZ nanofibers exhibit relatively smooth surfaces and homogeneous one-dimensional structures with diameters ranging from 200 to 250 nm. As shown in Figure 3c,d, the diameters of nanofibers range from 150 to 200 nm after calcination at 500 °C due to the removal of PVP. After loading the C nanostructures onto the TZ nanofibers, the nanofibers were no longer smooth. The C nanospheres were uniformly loaded onto each nanofiber without aggregation, which indicated that a hierarchical heterogeneous structure was successfully constructed, as shown in Figure 3e. In addition, the EDS spectra from the corresponding SEM images indicate that the TZC nanofiber filmare composed of O, Ti, C and Zn elements, as shown in Figure 3f. To strengthen the evidence, elemental mappings were investigated to identify the spatial distributions of O, Ti, C and Zn in the heterojunction, as shown in Figure 3g,h. Obviously, Zn and C were uniformly distributed in the composite nanofiber film, further indicating that C was successfully loaded onto the surface of TZ nanofibers. An HRTEM image was used to further study the microstructure information of the TZC heterojunction. As shown in Figure 3i, crystal lattice fringes with d-spacing of 0.35 and 0.28 nm in HRTEM image are assigned to the lattice plane of anatase $TiO_2$ (101) and ZnO (100).

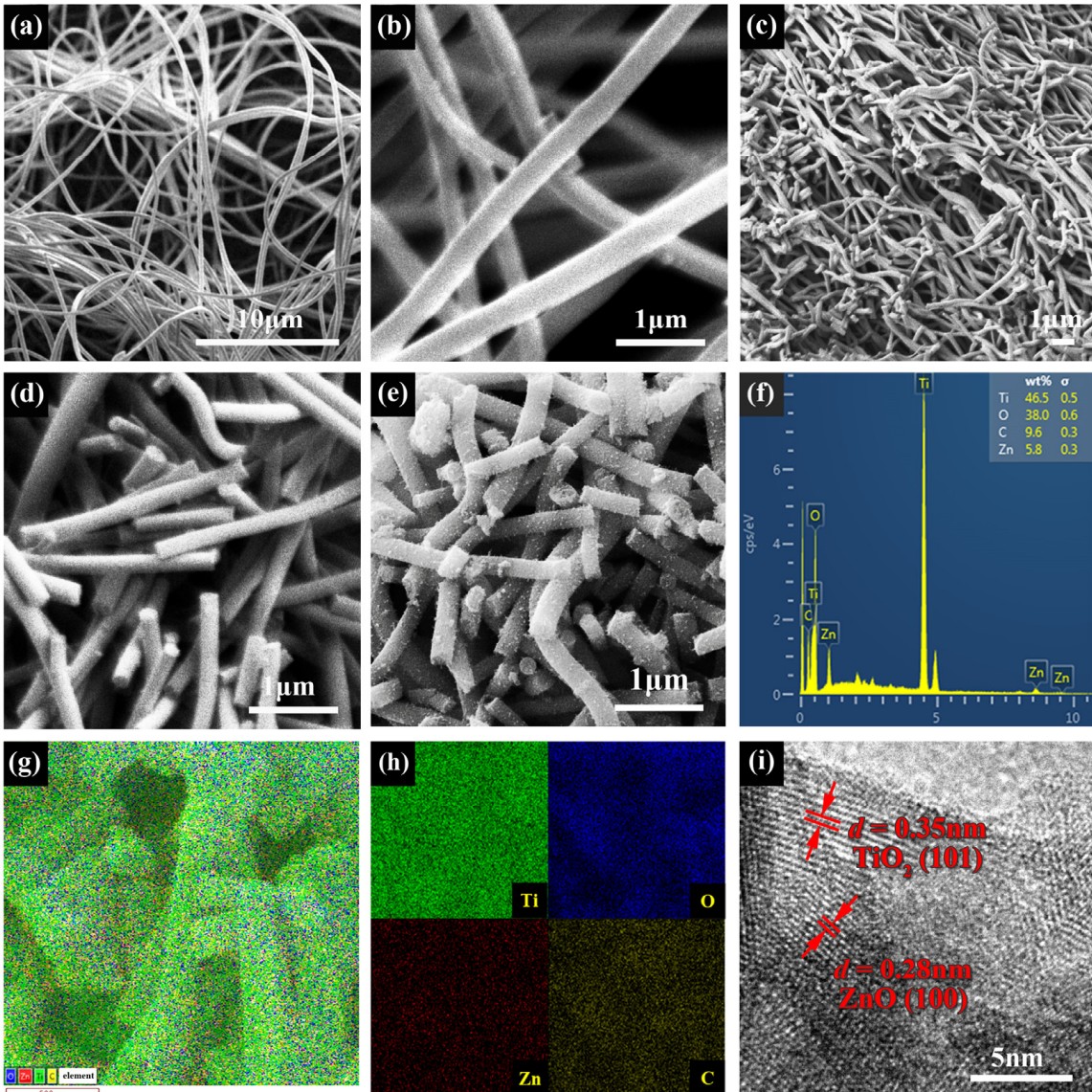

**Figure 3.** SEM images of (**a**) PVP/TZ; (**b**) PVP/TZ enlarged; (**c**) TZ;(**d**) TZ enlarged; (**e**) TZC; (**f**) EDS spectra of TZC; (**g**) EDS mapping of TZC; (**h**) elemental distribution; (**i**) HRTEM image of TZC.

Figure 4 presents the XRD patterns. All samples reveal a typical anatase $TiO_2$ (JCPDS 73-1764) with obvious characteristic peaks at approximately 25.3°, 38.1°, 48.1°, 54.6° and 62.9°, which are ascribed to (101), (004), (200), (105) and (204) crystal planes, respectively. All the diffraction peaks are sharp, indicating excellent crystallinity because of the proper calcination temperature. Moreover, a clear indication of the presence of ZnO (JCPDS 36-1451) comes from two peaks located at 31.1° and 36.3°, ascribable to (100) and (101) crystal planes in the TZ and TZC samples, respectively. Due to the low content of ZnO, the ZnO peaks are weaker than the $TiO_2$ peaks. This indicates that a part of Zn formed in the ZnO crystal structure instead of entering the $TiO_2$ lattice, and the ZnO crystal is tightly attached to the $TiO_2$ octahedral crystal both internally and on the surface with PVP removal during calcination. As shown in Figure 4b, the X-ray diffraction peaks of the an atase crystal plane (101) in TZ and TZC are slightly shifted toward a lower diffraction angle with the Zn-doped $TiO_2$ compared to T0. This finding suggests that similar ionic radii of dopant Zn (the radius of the $Zn^{2+}$ ion in the oxide is 0.074 nm) and $TiO_2$ (the radius of the $Ti^{4+}$ ion in the oxide is 0.068 nm) led to lattice distortion of the anatase structure by another part of Zn entering the lattice of $TiO_2$ [37,38]. These two ways of Zn entering the $TiO_2$ nanofibers

are present simultaneously. Crystallite size (*D*), dislocation density (*ρ*) and lattice strain (*ε*) were calculated by the following equations [39]:

$$D = K\lambda/(\beta\cos\theta), \qquad (1)$$

$$\rho = 1/D^2, \qquad (2)$$

$$\varepsilon = \beta\cos\theta/4, \qquad (3)$$

where *λ* is the X-ray wavelength, *β* is FWHM, *θ* is the diffraction angle, *K* is a constant 0.9. Table 2 shows a comparison of the crystal microstructures. C and Zn entering the TiO$_2$ lattice changed microstructure, with TZC being the most pronounced. The increased *ρ* and *ε* make the crystals more disordered and more defects, which helps to improve the photocatalytic efficiency. No characteristic peaks attributed to C were detected.

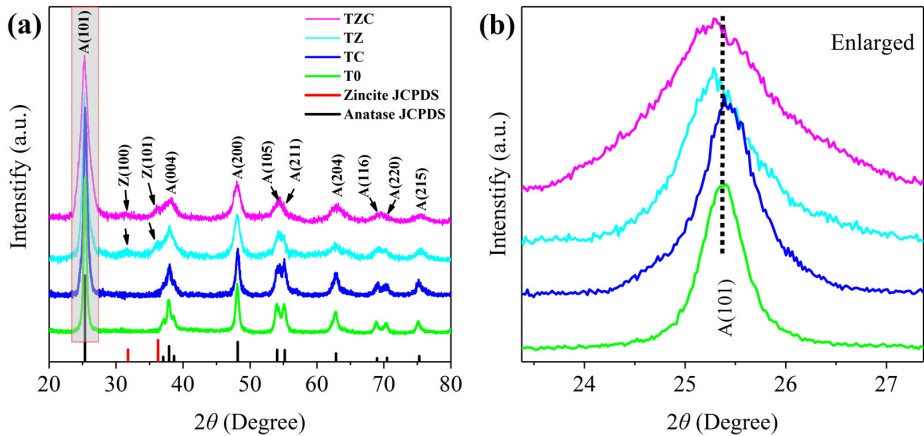

**Figure 4.** (**a**) XRD patterns of all samples; (**b**) crystal plane (101) enlargedimage.

**Table 2.** Comparison of microstructure of all samples.

| Samples | *D* (nm) | *ρ* (1/nm)$^2$ | *E* |
|---------|----------|----------------|-----|
| T0 | 19 | 0.003 | 0.0018 |
| TC | 14 | 0.005 | 0.0024 |
| TZ | 10 | 0.009 | 0.0034 |
| TZC | 7 | 0.02 | 0.0050 |

Raman spectroscopy is the process of interaction between light and chemical bonds inside materials, which is closely related to the vibrational energy level of materials, especially nanomaterials. Each spectral peak corresponds to a specific chemical bond vibration, which corresponds to information about the material crystal concentration, structure, deformation and defects.

Figure 5a shows the Raman of all nanofiber films. The typical Raman shifts of TiO$_2$ are obtained in all the samples. The Raman shifts at approximately 147 cm$^{-1}$, 399 cm$^{-1}$, 519 cm$^{-1}$ and 637 cm$^{-1}$ are attributed to $E_g$, $B_{1g}$, $A_{1g} + B_{1g}$, and $E_g$ modes of the anatase TiO$_2$, respectively [40–42]. The $A_{1g} + B_{1g}$ mode peak at 519 cm$^{-1}$ and the $E_g$ mode peak at 147 cm$^{-1}$ are caused by the Ti-O bond bending and stretching vibrations, respectively [43]. $A_{1g} + B_{1g}$ mode peaks of C-modified TC and TZC are significantly weaker compared to T0 and TZ, which indicates that the bending vibrations of Ti-O bond are altered due to the entry of C into the TiO$_2$ lattice. To further explore the effect of C and Zn on the Raman spectra of TiO$_2$ nanofibers, the strongest peaks are enlarged in Figure 5b. It is seen that the strongest peak positions of TC and TZC are slightly shifted to a higher wavenumber compared with T0 and TZ. The shift of the vibrational peaks indicated that C was successfully loaded onto TiO$_2$ and entered the lattice, leading to distortion. The

weakening of the TC and TZC vibrational peaks is attributed to the black C covering the surface of nanofibers, blocking the Raman scattering of the laser. The increased FWHM of TC and TZC vibrational peaks indicates that the loading of C leads to poor crystallinity of anatase and suitable crystal defects, which are beneficial for photocatalytic reactions. At the peak, 1599 cm$^{-1}$ is attributed to the G-band of graphite exhibiting ordered carbon atoms in TC and TZC samples [44,45]. The G-band is formed by the stretching vibration of all sp2 atoms in the carbon ring or long chain under incident light. This indicates that the presence of a large amount of C in the form of graphite on the surface of the nanofibers can improve the conductivity of the TiO$_2$ surface, thus facilitating the transfer of photogenerated carriers to the organic dye surface. The D-band representing the disordered C is not found. This does not represent the absence of disordered graphite. It has been reported that the D-band decreases in intensity with increasing energy of the incident light wavelength [46]. The D-band may not be detected when the graphite content is relatively low, and the higher energy 325 nm wavelength is used in the test. The above analysis illustrates the existence of TiO$_2$ and C. Unfortunately, there was no peak of ZnO, probably due to the low doping content or the blocking of surface C, but the construction of ZnO was successfully observed in XRD and TEM.

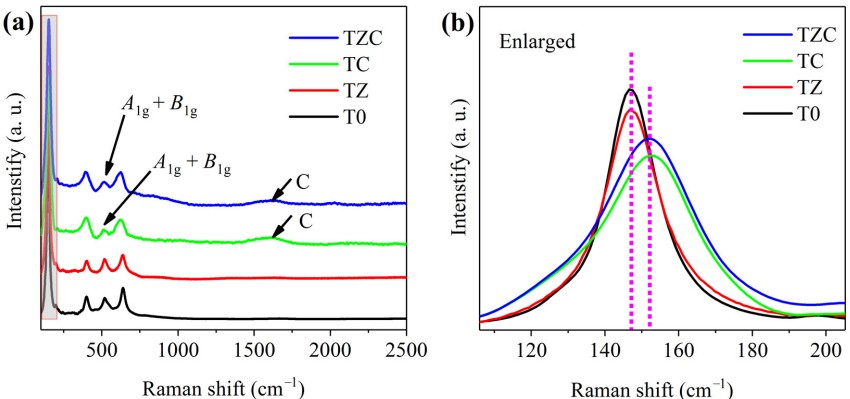

**Figure 5.** (**a**) Raman spectra of all samples; (**b**) enlarged image of the strongest peaks for all samples.

XPS was applied to elucidate the presence of elements and their surface electronic states on TZC nanofiber film. The full survey of XPS spectra shows that TZC nanofibers are composed of Ti, Zn, C and O elements, consistent with the EDS, as shown in Figure 6a. Although XPS was a semi-quantitative measurement, the percent content of the element was calculated based on the peak area and the sensitivity impact factor of the instrument. The atomic percentages of C, O, Zn and Ti were calculated from the C$_{1s}$, O$_{1s}$, Zn$_{2p}$ and Ti$_{2p}$ curves to be 28.33%, 48.29%, 3.33% and 20.05%, respectively. C blocks a part of the surface of the TiO$_2$ nanofibers exposed to X-rays, resulting in a lower content of Ti and a higher content of C in the sample. The sample is easily contaminated by C of air, due to the way XPS works. To estimate the percentage of modified C, the percentage of atomic number of C in EDS was calculated as 18.89%. Thus, C from the air accounts for 9.44%. In Figure 6b, the high-resolution spectra of Ti$_{2p}$ shows peaks at 471.6, 464.2 and458.3 eV, corresponding to Ti$_{2p}$ satellite, Ti$_{2p1/2}$ and Ti$_{2p3/2}$, respectively [47,48]. The inter-peak distances of Ti$_{2p1/2}$ and Ti$_{2p3/2}$ at 5.9 eV indicates that the element Ti is present in the nanofibers in the form of crystals of TiO$_2$ [49]. To fit the Gaussian components perfectly, the broad Ti$_{2p3/2}$ and Ti$_{2p1/2}$ peaks were decomposed into two peaks separated by about 1 eV. The main peaks at 458.5 eV and 464.4 eV are attributed to ion Ti$^{4+}$, while the minor peaks at 457.7 eV and 463.2 eV are assigned to ion Ti$^{3+}$ [50,51]. The presence of a few ionic Ti$^{3+}$ molecules on the surface of TiO$_2$ was reported; moreover, XPS analysis was consistent with the study. The presence of ionic Ti$^{3+}$ leads to weak absorption of visible light in the composite nanofibers, leading to absorption edge redshift (consistent with UV), which is beneficial for improving the photocatalytic efficiency [52]. In Figure 6c, the high-resolution spectra of Zn$_{2p}$ shows

peaks at 1021.3 and 1044.3 eV for $Zn_{2p3/2}$ and $Zn_{2p1/2}$, respectively, indicating the existence of a $Zn^{2+}$ oxidation state [53]. The high-resolution $C_{1s}$ spectra are decomposed into five peaks at 283.7, 284.8, 285.6, 286.8 and 288.7 eV, as shown in Figure 6d. XPS $C_{1s}$ shows that C interacts with $TiO_2$ in the form of chemical bonds. The peaks at 283.7 and 285.6 eV represent the Ti-C bond and Ti-O-C group, respectively, exhibiting the chemical bonds formed by C with other elements in $TiO_2$ [54,55]. The chemical bonds formed between C and the elements in $TiO_2$ are beneficial for the transfer of carriers between C and $TiO_2$ [56]. The surface C changes the lattice structure of $TiO_2$, while FS spectra are also observed. The peaks at 284.8, 286.8 and 288.7 eV are attributed to the adventitious carbon of C–C, C–O and C=O bonds, respectively [57]. In Figure 6e, the high-resolution $O_{1s}$ spectra indicates that the peaks at 529.7, 530.3, 531.4 and 532.3 eV are attributed to metal oxide, hydroxyl, C=O and C–O, respectively [58]. From metal oxide, itcan be inferred that $TiO_2$ and ZnO coexist as a heterostructure in TZC. Since the photodegradation efficiency of $TiO_2$ depended on the current carrier distribution at the valence band (VB) potential top and conduction band (CB) potential bottom, XPS valence band spectra were detected to further investigate the bandgap structure. The intersections of the tangents indicate that the VB potential of TZC is 2.186 eV, as shown in Figure 6f.

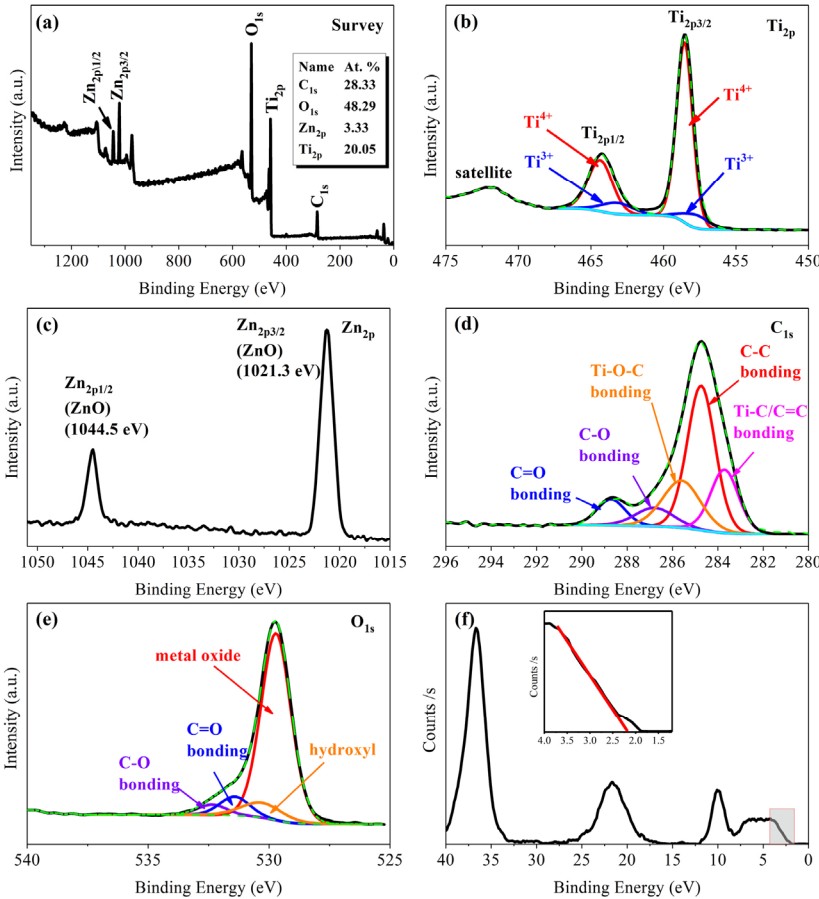

**Figure 6.** XPS spectra of TZC (**a**) survey; (**b**) $Ti_{2p}$; (**c**) $Zn_{2p}$; (**d**) $C_{1s}$; (**e**) $O_{1s}$; (**f**) valence.

UV-v is DRS is an effective means of studying the energy band structure of materials [59]. To investigate the optical absorption and heterojunction electronic structures, UV-v is DRS and Tauc's plots of all samples were applied. All samples exhibit a large slope of the absorption edge curve with a narrow absorption range. Compared with T0, the absorption edges of other samples of incorporated Zn and C are shifted toward a longer wavelength, as seen in Figure 7a. The absorption edge shift indicates charge transfer at the interfaces of$TiO_2$, ZnO and C. The absorption band edges of all samples range from

417 (T0) to 431 (TZC) nm. Figure 7b shows the Tauc's plots used to evaluate the band-gap energy. The bandgap of each photocatalyst composite is obtained from the intercept $hv$ axis by plotting the tangent line of the linear part of curve. The bandgapsare (3.210 ± 0.003), (3.190 ± 0.004), (3.101 ± 0.004) and (3.079 ± 0.003) eVfor T0, TZ, TC and TZC, respectively. The narrowing of the band gap due to Zn-doped and C-modified $TiO_2$ nanofibers results in a redshift. Redshift allows lower-energy light to be used for photocatalysis, thereby improving the photocatalytic activity of the photocatalyst. As a result, both modified C and ZnO contribute to the absorbance enhancement of TZC samples. The C-modified and Zn-doped $TiO_2$ nanofiber films also show a redshift in the absorption edges. Besides the effect of heterojunction, when C and Zn enter the $TiO_2$ lattice, causing crystal defects and lattice bending, the redshift of the absorption edge is also induced. Crystal defects are another important factor in reducing the band gap. The redshift of the absorption edge is the result of a combination of two factors. Subsequently, the conduction band (CB) potentials of TZC was calculated according to the following equations:

$$E_{CB} = E_{VB} - E_g, \tag{4}$$

where $E_{CB}$, $E_{VB}$ and $E_g$ are the conduction band potential, valence band potential and band gap, respectively [60]. According to the valence-band XPS spectra, the $E_{VB}$ values of TZC is 2.186 eV. Consequently, the $E_{CB}$ value is calculated to be (−0.893 ± 0.003) eV for TZC. The samples do not absorb significantly in the visible range, so the activation is borderline among UV and visible edge, and thus without a clear indication of the activation requirements.

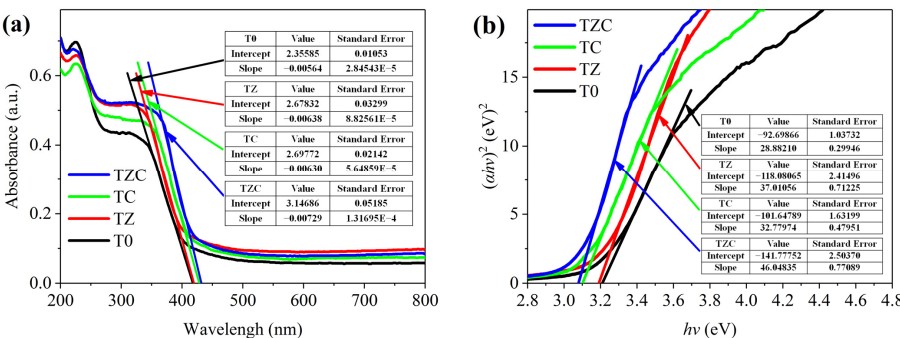

**Figure 7.** (**a**) UV-vis DRS; (**b**) Tauc's plots of all samples.

Fluorescence spectra (FS) reflect the lifetimes and recombination rates of separated photogenerated holes and electrons [61]. Figure 8 shows the FS of all samples with 300 nm excitation at room temperature. The stronger peaks at about 393 nm are attributed to photogenerated electron-hole pairs recombination in valence and conduction band [62,63]. FS are able to provide information on the oxygen vacancies of the $TiO_2$ composites and the surface states of the nanofibers [64]. Two weak peaks at 463 and 477 nm are attributed to oxygen vacancy associated with $Ti^{3+}$ centers and oxygen vacancy with trapped electrons or band edge emission [65]. Moreover, after the ion $Zn^{2+}$ enters the lattice to replace the ion $Ti^{4+}$, oxygen vacancies are formed due to the difference in the valence, leading to the absence of oxygen shedding. The peaks of TZ and TZC at 393 nm are stronger than the others compared with T0 and TC, indicating the presence of ZnO. The UV luminescence peak of ZnO is at around 391 nm [66]. A shoulder appears at 423 nm in the ZC and ZTC samples, which may belong to the effect produced by the C entry into the lattice. FS intensity of all samples is attenuated compared to T0, indicating the improvement of the separation and recombination efficiency of photogenerated electron-hole pairs by heterostructure energy level adjustment strategy. When heterojunction structures and C modifications are presented on the surface of the nanofiber film, they change the luminescence characteristics of $TiO_2$. Thus, the TZC heterostructure and crystal defects enhance the photocatalytic activity of the nanofiber film.

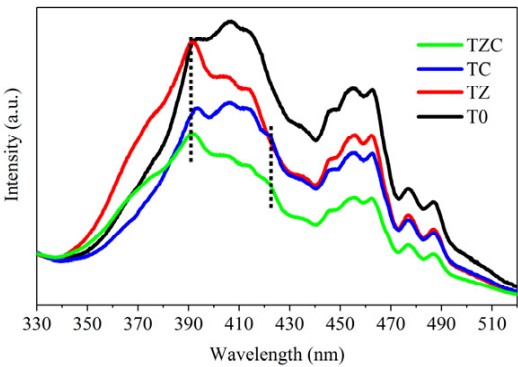

**Figure 8.** Fluorescence spectra (FS) of all samples.

As shown in Figure 9, the photodegradation activities of all samples were examined by degradation of simulated pollutant dyes (MB, MO, RhB, MG) in aqueous suspension under simulated sunlight. To investigate the role of photocatalysts, degradation experiments of organic dyes MB, MO, RhB and MG without photocatalysts were tested under light, defined as MB0, MO0, RhB0 and MG0, respectively. As a comparison, the results of the blank experiments showed no significant degradation of all dyes in the absence of photocatalyst, indicating that all dyes were degraded by the photocatalytic reaction of the photocatalyst. $C/C_0$ was used to calculate the photocatalytic efficiency [67]. All dyes' degradation efficiencies of T0 are very low, only 52.1%, 55.6%, 50.7% and 72.1% for MB (Figure 9a), MO (Figure 9b), RhB (Figure 9c) and MG (Figure 9d) with 70 min, 70 min, 90 min and 25 min reaction, respectively. No degradation of any of the dyes was observed when light or photocatalyst was absent. This indicates that all degradation is attributed to the photocatalytic reaction. The degradation efficiency increases with the entry of C and Zn. TZC exhibits the highest efficiency of degradation with 90.3%, 85.6%, 87.2% and 90.6% for MB, MO, RhB and MG with 70 min, 70 min, 90 min and 15 min reactions, respectively. Among the four dyes, MG was the easiest to degrade and RhB was the most difficult. The photocatalytic degradation of organic pollutant dyes conforms the pseudo-first-order reaction kinetic equation:

$$\ln (C_0/C) = kt, \tag{5}$$

where $C_0$ and $C$ are the initial concentration and the concentration at $t$ of simulated pollutants, respectively; $k$ is the rate constant; $t$ is the reaction time [68]. The corresponding pseudo-first-order reaction kinetic fitted slope lines are displayed, with the highest rate constants for the degradation of all dyes at TZC, in Figure 10. In addition, $k$ values for degradation of all dyes were estimated from the kinetic fitted curves, as shown in Figure 11. To evaluate the standard error of the fitted $k$ values, error bars are added to confirm that the calculated parameters are within the error tolerance. In particular, the activity ($k$-values) of TZC is3.01, 3.01, 2.36 and 2.89 times as high as T0 for the degradation of MB, MG, MO and RhB, respectively. Notably, the corresponding results are very similar to the trend in the rate constant $k$. The above results confirm that the photocatalytic improvement of TZC is effective for all dyes. As shown in Figure 12, the reusability and stability of TZC was examined. There is no obvious decline after five cycles with the same catalyst, implying the high stability of TZC nanofiber film. The $k$ values in all cycling experiments were calculated in order to further evaluate the effect of cycling experiments on the photocatalyst activity in Figure 13. The $k$ values shows irregular fluctuations in all dyes, but the variations are in the acceptable range. No significant decrease in $k$ value is found in the cycling experiments, indicating that the photocatalysts have good stability and recyclability.

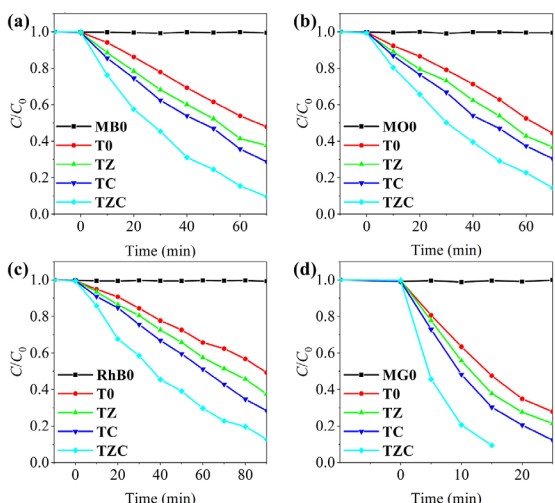

**Figure 9.** Photocatalytic degradation efficiency of (**a**) MB; (**b**) MO; (**c**) RhB; (**d**) MG.

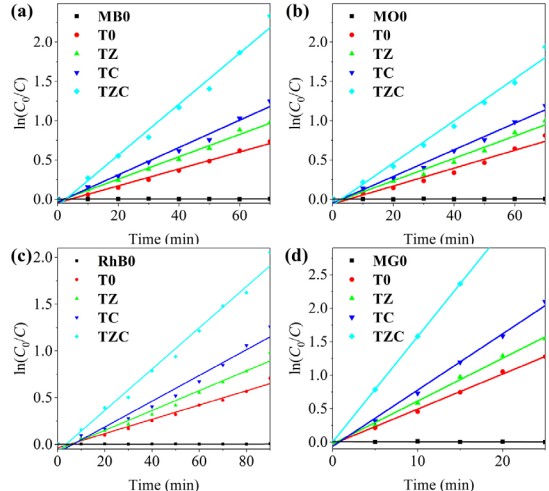

**Figure 10.** The corresponding pseudo-first-order reaction kinetic fitted curves (**a**) MB; (**b**) MO; (**c**) RhB; (**d**) MG.

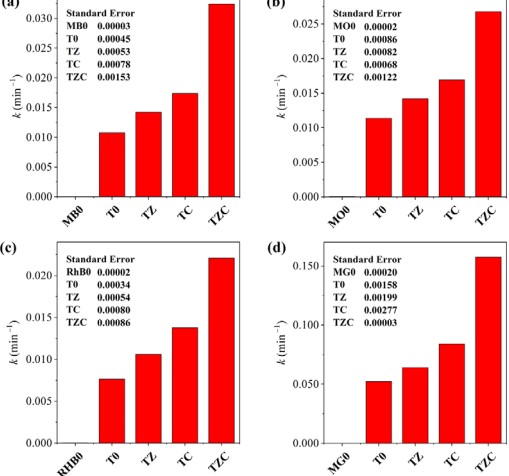

**Figure 11.** The kinetic values of all samples for the degradation of (**a**) MB; (**b**) MO; (**c**) RhB; (**d**) MG.

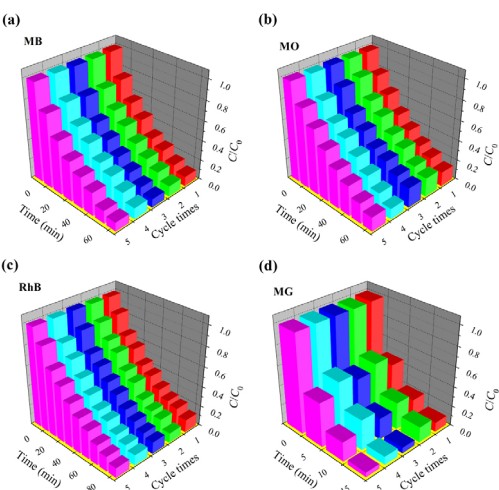

**Figure 12.** Recycling test of the TZC for the degradation of (**a**) MB; (**b**) MO; (**c**) RhB; (**d**) MG.

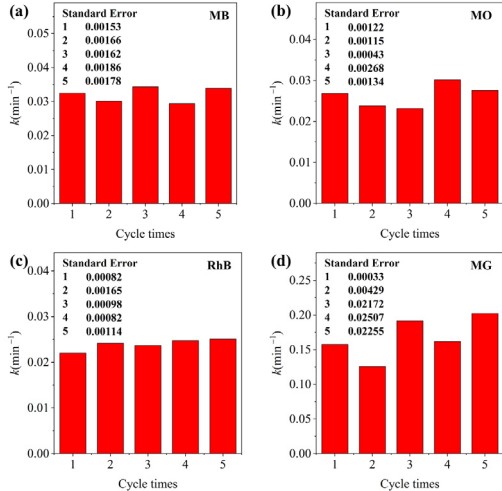

**Figure 13.** *k* of recycling test (**a**) MB; (**b**) MO; (**c**) RhB; (**d**) MG.

Figure 14 shows the band gap structure of C-modified Zn-doped $TiO_2$. When the heterojunction is constructed, to balance their Fermi levels, electrons are transferred from ZnO to $TiO_2$ through the interface, leaving positively charged holes, as shown in Figure 14a. As a result, a positively charged electron-depletion layer exists in ZnO and a negatively charged electron accumulation layer exists in $TiO_2$, generating an internal electric field from ZnO to $TiO_2$ at the interface, as shown in Figure 14b. The internal electric field causes band bending of $TiO_2$ and ZnO, as shown in Figure 14c. When $TiO_2$ and ZnO are irradiated by photons exceeding their band gap energy, photogenerated electrons are photoexcited from VB to CB, so the positively charged holes are retained. With the help of the internal electric field and the band bending, photogenerated electrons are transferred from $TiO_2$ to ZnO and holes are transferred from the ZnO to $TiO_2$, achieving the separation and transfer of photogenerated electron-hole pairs at the interface, as shown in Figure 14d. Due to the electron mobility and low band structure of C, photogenerated electrons are forced to move from CB of ZnOor $TiO_2$ to C, which effectively separates and hinders recombination of photogenerated electron-hole pairs during electron transport. To summarize, the enhanced photocatalytic activity is due to the heterojunction reducing the recombination of electron-hole pairs increasing the intensity and range of light absorption by black C.

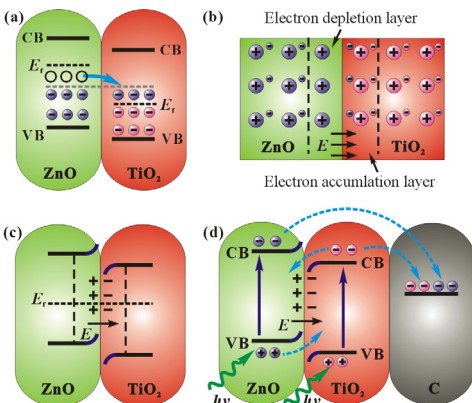

**Figure 14.** Schematic diagram of the possible mechanism for the charge-transfer process in the heterojunction (**a**) ZnO/TiO$_2$ heterojunction; (**b**) electric field at the interface of ZnO/Ti$_2$O; (**c**) band bending of ZnO and TiO$_2$; (**d**) ZnO/TiO$_2$/C heterojunction.

Figure 15 illustrates the possible TiO$_2$ crystal structure after the entry of Zn and C into the lattice. When Zn$^{2+}$ enters the TiO$_2$ lattice, oxygen vacancies appear due to the absence of oxygen. Subsequently, photogenerated electrons are transferred by occupying the oxygen vacancies, and thus, these oxygen vacancies become the transport pathway for photogenerated carriers. C is attached to the TiO$_2$ surface by chemical reaction and calcined at high temperature, which provides the opportunity for C to enter the surface lattice. Accompanied by the breakage of the original chemical bond of C during calcination, the thermal motion causes C to enter the TiO$_2$ lattice, possibly due to the close contact with TiO$_2$. After C enters the lattice, it may replace O to form Ti-C bonds or Ti to form covalent bonds with the surrounding oxygen. This effect occurs only on the TiO$_2$ surface because C does not have enough energy to enter the inside of the crystal to form a uniform doping like Zn. This defect helps photogenerated electrons to reach the surface more easily to participate in the photocatalytic reaction. Heterojunction and doping simultaneously change the crystal structure of TiO$_2$, thus improving the photocatalytic efficiency.

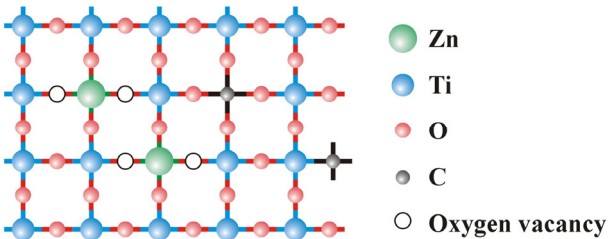

**Figure 15.** Schematic diagram of possible crystal defects.

## 4. Conclusions

We have designed and synthesized a novel ternary photocatalytic material of C-modified Zn-dopedTiO$_2$ ternary composite nanofiber film by electrospinning and hydrothermal methods. The optimized ternary composite photocatalyst exhibited the highest degradation efficiency for all organic pollutant dyes. The heterojunction structure among ZnO, TiO$_2$ and C reduce the recombination of electron-hole pairs while providing a pathway for photogenerated electron separation. Moreover, nanofibers have a large specific surface area compared to bulk materials and crystal defects, which benefits its interfacial contact with organic pollutants and photogenerated electron transport. Nanofibers have a faster settling speed and are easily recycled as a small-area film material compared to nanoparticles. Therefore, C-modified Zn-dopedTiO$_2$ composite nanofiber film may be a good candidate for industry photocatalysis application due to its advantages of separation freedom, efficiency and stability.

**Author Contributions:** Conceptualization, Y.L. and J.H.; methodology, Y.L.; formal analysis, Y.L.; investigation, J.H.; writing—original draft preparation, Y.L.; writing—review and editing, Y.L. and X.Q.; supervision, X.Q.; project administration, Y.L. All authors have read and agreed to the published version of the manuscript.

**Funding:** This research was funded by the Basic Science Research Foundation of Heilongjiang Province, grant number 2019-KYYWF-1397.

**Institutional Review Board Statement:** Not applicable.

**Informed Consent Statement:** Not applicable.

**Data Availability Statement:** Not applicable.

**Conflicts of Interest:** The authors declare no conflict of interest.

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
