# Peer review of "Synergistic Effects of Multiple Heterojunctions and Dopant Atom for Enhancing the Photocatalytic Activity of C-Modified Zn-Doped TiO2 Nanofiber Film"

_coatings, doi:10.3390/coatings13030647_

Round 1
Reviewer 1 Report
Authors need to address the following before accepted for the publication.
1. Figure 4 - Please assign the peaks in the XRD patterns
2. Figure 6 - X axis of the XPS plots are organized in the reverse order....ex- Figure 6 (a) 1400-0....Please change all the figures accordingly.
3. Figure 6 - 2p3/2, 2p1/2...spin should be subscripted
4. Figure 7 - Absorbance vs. wavelength and tauc plots...extrapolations should be done for all the four graphs
5. Figure 9 and 10 - Figures need to be more resolved....Use big font...thck lines...
6. Figure 2 - What are these red and black dots...they need to labeled properly
7. Authors may cite the following to improve the quality of the manucript.
Photocatalytic activity of Fe and Cu co-doped TiO2 nanoparticles under visible light.
Photocatalytic activity of N, Fe and Cu co-doped TiO2 nanoparticles under sunlight
Activity enhanced TiO2 nanomaterials for photodegradation of dyes
Efficient photocatalysis of carbon coupled TiO2 to degrade pollutants in wastewater – A review
Reviewer 2 Report
I need some clarification, authors not expanded the C even in the title or introduction, for common readers you mention Carbon modified
What type of carbon whether it is in the form of any crystalline nature.
In the end of introduction you have to mention the Scope of the research, what are you going to do.
Carbon is directly inserted into the lattice, not anchored in the lattice of TiO2, explain how it is directly inserted ... what is the driving force.
In Flurecesen spectra what is the difference after Carbon modification, what type of Carbon structure exits in the final film, from Raman spectra D-band identified
Author Response
Thank you for your careful review comments, which made the article even better. I apologize for taking up a lot of your time with too many faults in the manuscript.
Point 1: Figure 4 - Please assign the peaks in the XRD patterns
Response 1: Thanks for your reviewer. I assigned the peaks in the XRD patterns. A screenshot of the revised manuscript is as follows.
Point 2: Figure 6 - X axis of the XPS plots are organized in the reverse order....ex- Figure 6 (a) 1400-0....Please change all the figures accordingly.
Point 3: Figure 6 - 2p3/2, 2p1/2...spin should be subscripted.
Response 2,3: Corrections were made according to the reviewer. Subscripts are used in the manuscript. A screenshot of the revised manuscript is as follows.
Point 4: Figure 7 - Absorbance vs. wavelength and tauc plots...extrapolations should be done for all the four graphs
Response 4: According to your the reviewer, Figure (a) was extrapolated to make the graph more complete. However, Figure (b) was only moderately extrapolated as the invalid straight line was too long for the reader to see the change in slope of the absorption edge if it was a complete graph. A screenshot of the revised manuscript is as follows.
Point 5: Figure 9 and 10 - Figures need to be more resolved....Use big font...thck lines...
Response 5: Figure 9 and 10 were used big font and thck lines. A screenshot of the revised manuscript is as follows.
Point 6: Figure 2 - What are these red and black dots...they need to labeled properly
Response 6: Figure 2 - The red and black dots were labeled. A screenshot of the revised manuscript is as follows.
Point 7: Authors may cite the following to improve the quality of the manucript.
Response 7: Thank you for your recommendation, we cite these excellent references. A screenshot of the revised manuscript is as follows.
Reviewer 3 Report
Comments:
In the submitted paper the authors describe the synthesis and an extended characterization of Zn-TiO2-C nanofibers. The use of such compounds as photocatalysts with respect to several dye molecules degradation is also investigated and discussed. The paper appears interesting and substantially well arranged in most of its part, nevertheless some inaccuracies need to be addressed before its publication on Coatings. In the present state the paper need a major revision.
In the following comments (not in priority order) I resume the main issues to reconsider in order to improve the manuscript.
1) In my opinion, the title appears as a sentence rather than a very title.
2) The authors should better highlight the novelty of their work and the improvement with respect to the state of the art.
3) The authors should better explain the nature of their compounds. In the abstract it is reported that C and Zn-doped TiO2 is prepared but in the description and the results discussion it is clear that Zn is properly a dopant (enters into TiO2 crystal lattice) and constitutes ZnO phase at the same.
4) Row 92: chemical formula of zinc acetate.
5) Row 105: please also specify the treatment time.
6) In my opinion the scheme of figure 2 is not very clear. Red dots become black dots… Authors could add reactants, reactions and the occurred treatments over the arrows.
7) Rows 137 and 151: I suppose Shimadzu.
8) Concerning synthesis, authors could make clear the final weight ratios among ZnO-TiO2-C.
9) Section 2.5. The authors used 1 g/L photocatalyst loading: did they evaluate different loadings? Why?
The authors used 10 minutes for reaching adsorption/desorption equilibrium. Is this duration enough? (usually 30 minutes are used).
The pH is given at 7. Was it reached by tampons?
The temperature is set at 30°C: why this choice? It appears quite high with respect to 20-25°C usually employed. Why?
10) SEM images 3a and 3b are not very in focus in my opinion. Maybe some charging effects affects the measurements (images 3c, with carbon, appears better); did the authors try the metallization of the sample? Why?
The legend in figure 3e is not very readable. Moreover, I suggest the use of different colours for Zn and C (with higher contrast with respect to the used ones) because the elements are not visible in the map.
11) Figure 4a. Zincite and anatase are not XRD patterns but just JCPDS references.
Concerning XRD measurements, crystallite dimensions could be evaluated and discussed.
12) Raman peak at 198 cm-1 is not visible at all.
13) Row 220-221. Defects are double-faced points: they also act as recombination points.
14) The XPS composition is given with 2 decimal places. Is this correct? (In my knowledge the sensitivity for XPS is 1% or, to the outmost, 0.1%)
Concerning XPS measurements, only the TZC outcomes are presented and discussed. What about the other three compounds? What about their carbon content? I mean, 25-30% at. for carbon is quite common also for not carbon containing compounds. Again on carbon, authors deconvoluted the C signal into several components: can they quantify the “useful” carbon with respect to the adventitious one? Concerning oxygen discussion, authors did not consider and attributed any signal to hydroxyls: why?
Figure 6 caption: TZR10 is reported. I suppose it is TZC. Moreover, figure 6f is written as 6d.
15) Row 269-270: “All samples exhibited the sharp absorption.” Please comment this sentence.
16) Equation 1 (row 287): please control it.
17) Row 305: I suggest the use of “shoulder” instead of “peak”.
18) Figure 9-10-11: please better describe in caption and text the meaning of MB0, MO0, etc.
In figure 9 BM is written instead of MB.
19) Except MG (figure 9d), MB, MO and RhB do not show the typical first-order decay; they decrease linearly. This is probably due to the low photo-activity of the catalysts (except TZC).
20) Data in figure 11 are given without error bars, please add them.
21) I suggest to show and comment also the UV-Vis spectra trends for the four dyes. From the spectra, in fact, it possible to understand if the solutions are simply bleached or if the dyes are further degraded. As a general consideration, when an improved light absorption is invoked for justify the improvement of a catalyst performance (rows 365-366) and a Xe lamp is used (both UV and visible emission) for the process activation, further investigations are needed for a proper evaluation of the UV and visible range contribution, especially when dyes are used (i.e. visible absorbing molecules). Maybe a not coloured probe should be tested.
22) Figure 12 is not so useful in my opinion. The comparison of kinetic constants in the recycling test are more interesting.
23) Concerning the catalysts recovery, authors completely skipped the recovery step: how did they do it? What about the used time? What about the recovered amount?
24) Row 389-390: what about the specific surface area of the prepared samples? Authors correctly underlined the characteristics of the fibres and the surface area data should be added.
Author Response
Thank you for your careful review comments, which made the article even better. I apologize for taking up a lot of your time with too many faults in the manuscript.
Point 1: In my opinion, the title appears as a sentence rather than a very title.
Response 1: Thanks for the advice, this is indeed a problem. Language barriers have always plagued non-English speaking authors. We revised the title of the paper to be more suitable. A screenshot of the revised manuscript is as follows. A screenshot of the revised manuscript is as follows.
Point 2: The authors should better highlight the novelty of their work and the improvement with respect to the state of the art.
Response 2: We revised the abstract, introduction and mechanism of the manuscript in the hope of better highlighting the novelty of the work and the improvements with respect to the state of the art. Screenshots of the revised manuscript is as follows.
Abstract
Introduction
mechanism
Point 3: The authors should better explain the nature of their compounds. In the abstract it is reported that C and Zn-doped TiO2 is prepared but in the description and the results discussion it is clear that Zn is properly a dopant (enters into TiO2 crystal lattice) and constitutes ZnO phase at the same.
Response 3: The descriptions of ZnO and graphite crystals have been added in the abstract to make the chemical phases clearer.
Point 4: Row 92: chemical formula of zinc acetate.
Response 4: Thank you for your care and professionalism. The error has been corrected. A screenshot of the revised manuscript is as follows.
Point 5: Row 105: please also specify the treatment time.
Response 5: We did not record the sample cooling time. When the crucible containing the samples was completely cooled, the samples were collected.
Point 6: In my opinion the scheme of figure 2 is not very clear. Red dots become black dots… Authors could add reactants, reactions and the occurred treatments over the arrows.
Response 6: Figure 2 was modified so that the design ideas were expressed more clearly. The changes that occur in each process are added. A screenshot of the revised manuscript is as follows.
Point 7: Rows 137 and 151: I suppose Shimadzu.
Response 7: Thank you for your careful review. The error has been corrected. Screenshots of the revised manuscript is as follows.
Point 8: Concerning synthesis, authors could make clear the final weight ratios among ZnO-TiO2-C.
Response 8: C is adhered to the nanofiber surface by calcination. The loading rate of C can only be fixed by giving all the reaction conditions. The final amount of C cannot be calculated because the loading rate cannot be determined.
Point 9: Section 2.5. The authors used 1 g/L photocatalyst loading: did they evaluate different loadings? Why?
The authors used 10 minutes for reaching adsorption/desorption equilibrium. Is this duration enough? (usually 30 minutes are used).
The pH is given at 7. Was it reached by tampons?
The temperature is set at 30°C: why this choice? It appears quite high with respect to 20-25°C usually employed. Why?
Response 9: We did try different amounts of photocatalyst loading. When too little photocatalyst was loaded, the photocatalyst would fail completely and photocatalysis would not be possible. When there is too much photocatalysis, the photocatalysis is too fast to collect pollutants for evaluation. However, we did not focus our work on testing the loaded catalyst mass, so the testable range was not recorded. Such a ratio is only experience.
We also tried longer adsorption times, but no significant adsorption was observed. The experimental material was not porous and no significant adsorption was found.
Please forgive us for removing the expression that "the PH value is 7". This is a mistake in writing. Thank you very much for asking the question and giving us the opportunity to discover this almost inexcusable error. Since acetic acid was used during the hydrothermal process, the PH value was tested for concern about acetic acid residue during washing with water. PH of 7 is the wash water of the catalyst. The pH was not specifically adjusted for the simulated contaminants in the experiment. No PH testing was done either. As a writer, I sincerely apologize.
The tests were done in the summer and the lab did not have air conditioner to bring the temperature down to 20-25 °C. We don't have better experimental conditions.
Point 10: SEM images 3a and 3b are not very in focus in my opinion. Maybe some charging effects affects the measurements (images 3c, with carbon, appears better); did the authors try the metallization of the sample? Why?
The legend in figure 3e is not very readable. Moreover, I suggest the use of different colours for Zn and C (with higher contrast with respect to the used ones) because the elements are not visible in the map.
Response 10: All samples were coated with Pt prior to testing to increase conductivity. But the photo is still not clear. Possible causes are the instrument or the operator. It is very unfortunate that the SEM we received was not very high resolution. The samples were not recovered after testing, so they could not be tested again. We added small magnifications to SEM photos, while the photo brightness and contrast were adjusted in the hope of improving the clarity of the photos. It is expected that these measures will improve the reader's experience and help the reader to have a more comprehensive impression of material morphology.
Your advice is very useful, but the color of the element cannot be replaced without redoing the test. The brightness and contrast of the map are adjusted as much as possible and the element image is moved out so that the map has a larger area. Hopefully, such adjustment will improve the image quality. A screenshot of the revised manuscript is as follows.
Point 11: Figure 4a. Zincite and anatase are not XRD patterns but just JCPDS references.
Concerning XRD measurements, crystallite dimensions could be evaluated and discussed.
Response 11: The error has been fixed, and JCPDS has been added after Zincite and anatase. A discussion of crystallite size and dislocations has been added. Screenshots of the revised manuscript is as follows.
Point 12: Raman peak at 198 cm-1 is not visible at all.
Response 12: The 198 cm-1 peak is indeed too small to see clearly without enlargement and has been removed from the expression in the manuscript. A screenshot of the revised manuscript is as follows.
Point 13: Row 220-221. Defects are double-faced points: they also act as recombination points.
Response 13: I agree with your point about defects. Both too many and too few crystal defects are detrimental to the photocatalytic efficiency. The modifier for crystal defects was modified to make the presentation more accurate. A screenshot of the revised manuscript is as follows.
Point 14: The XPS composition is given with 2 decimal places. Is this correct? (In my knowledge the sensitivity for XPS is 1% or, to the outmost, 0.1%)
Concerning XPS measurements, only the TZC outcomes are presented and discussed. What about the other three compounds? What about their carbon content? I mean, 25-30% at. for carbon is quite common also for not carbon containing compounds. Again on carbon, authors deconvoluted the C signal into several components: can they quantify the “useful” carbon with respect to the adventitious one? Concerning oxygen discussion, authors did not consider and attributed any signal to hydroxyls: why?
Figure 6 caption: TZR10 is reported. I suppose it is TZC. Moreover, figure 6f is written as 6d.
Response 14: The XPS data sent back by the testing agency has only two decimal places, probably because the instrument is different.
XPS was sent to a commercial testing company for testing, and no other samples were done due to funding constraints. The impact of Additional C is unavoidable. XPS is a semi-quantitative instrument and the data obtained are estimates. The presence of carbon can be referred to both SEM and Raman, which contributes to a comprehensive understanding of the material composition. I don't know how to divide the C signal into useful carbons and perform peak splitting. No relevant literature or specific practices were identified. The limited time available to revise the manuscript has affected further research on this section. It is not clear to me whether the material that has been calcined at high temperatures contains hydroxyl groups. No references were searched for XPS analysis of hydroxyl radicals with oxide inorganic materials. I think the hydroxyl group may decompose and leave the material after high temperature calcination.
The title of Figure 6 was misspelled and has been corrected. A screenshot of the revised manuscript is as follows.
Point 15: Row 269-270: “All samples exhibited the sharp absorption.” Please comment this sentence.
Response 15: I think it was the English expression that appeared inappropriate. Perhaps it would be better to write " All samples exhibited a narrow absorption band with a large slope of the absorption edge curve." A screenshot of the revised manuscript is as follows.
Point 16: Equation 1 (row 287): please control it.
Response 16: The Equation has been written using the formula editor and been controlled. A screenshot of the revised manuscript is as follows.
Point 17:Row 305: I suggest the use of “shoulder” instead of “peak”.
Response 17: I will gladly accept your suggestion. Already replaced peak with shoulder. A screenshot of the revised manuscript is as follows.
Point 18: Figure 9-10-11: please better describe in caption and text the meaning of MB0, MO0, etc.
In figure 9 BM is written instead of MB.
Response 18: MB0, MO0, RhB0, MG0 are defined in text. Spelling errors were corrected. Screenshots of the revised manuscript is as follows.
Point 19:Except MG (figure 9d), MB, MO and RhB do not show the typical first-order decay; they decrease linearly. This is probably due to the low photo-activity of the catalysts (except TZC).
Response 19: I agree with your point of view. The UV-vis shows that the sample is unlikely to absorb visible light. The spherical xenon lamp used in the experiment does not have a large proportion of UV light. In addition, 50 ml beaker was used for the experiment and light was shone down through the mouth of the beaker. Organic dyes have a relatively small area to receive light. This may be the reason for the seemingly low photocatalyst activity. Fortunately, different kinds of photocatalyst comparison experiments were done to test the composite versus pure TiO2 photocatalytic activity under the same experimental conditions. It can be determined that the photocatalytic activity of the composites is higher than that of pure TiO2. The design of the composite material is successful.
Point 20: Data in figure 11 are given without error bars, please add them.
Response 20: The error bars are added in Figure 11. A screenshot of the revised manuscript is as follows.
Point 21: I suggest to show and comment also the UV-Vis spectra trends for the four dyes. From the spectra, in fact, it possible to understand if the solutions are simply bleached or if the dyes are further degraded. As a general consideration, when an improved light absorption is invoked for justify the improvement of a catalyst performance (rows 365-366) and a Xe lamp is used (both UV and visible emission) for the process activation, further investigations are needed for a proper evaluation of the UV and visible range contribution, especially when dyes are used (i.e. visible absorbing molecules). Maybe a not coloured probe should be tested.
Response 21: Unfortunately the contribution of UV and visible light in the photocatalytic process was not evaluated. The experiment was originally designed with the hope of using only sunlight, rather than additional artificial light sources, when dealing with water pollution. Artificial light sources may have more economic costs. The luminous spectrum of spherical xenon lamp is closest to sunlight, which is a good sunlight simulating light source. Please excuse two reasons for not being able to add to the mentioned experiments. One reason, purchasing UV lamps, filters and probes that are not available in the lab takes a long time in the mail, while the funding budget is overspent. Another reason, there was not enough time to complete additional experiments during the rewrite period. We will take your suggestions into full consideration when doing similar experiments in the future. The instruments needed for the experiment will be prepared before the experiment.
Point 22: Figure 12 is not so useful in my opinion. The comparison of kinetic constants in the recycling test are more interesting.
Response 22: The cyclic experiments took a lot of time, so please allow to keep the cyclic test images. I am also very interested in the calculation and comparison of the kinetic constants that should be used for cyclic experiments. It would be pleasant to have a reference.
Point 23: Concerning the catalysts recovery, authors completely skipped the recovery step: how did they do it? What about the used time? What about the recovered amount?
Response 23: The recovered samples consist of two parts. One part of the sample comes from the solution used to test the concentration. A small amount of sample is left in the solution during the extraction process and is recovered by centrifugation to ensure clarity of the solution. The photocatalyst needs to be separated from the solution to be measured quickly to prevent affecting the concentration of the organic solution. The other part comes from the degraded organic solution and the sample is recovered by precipitation. In fact the sample was deposited to the bottom of the beaker in 8-10 min, but a longer time was used for the experiment to reduce the loss of sample. Very hopelessly, the mass of the recovered samples was not weighed. A description of the recovery method has been added to the manuscript. A screenshot of the revised manuscript is as follows.
Point 24: Row 389-390: what about the specific surface area of the prepared samples? Authors correctly underlined the characteristics of the fibres and the surface area data should be added.
Response 24: Nanofiber films don't have a very large specific surface area like porous materials. Therefore, the experiment did not test the specific surface area. This part of the text was modified to make it more accurate. A screenshot of the revised manuscript is as follows.
Reviewer 4 Report
Organic dyes are harmful for animals and humans. The structural design of photocatalysts is an important method to improve photocatalytic activity.
I have reviewed the manuscript entitled:
„ Synergistic effects of multiple heterojunctions and dopant atom enhance the photocatalytic activity of C modified Zn-doped TiO2 ternary composite nanofiber film”.
In my opinion the manuscript need minor revision.
Comment1
Line 17 (MB, MO, RhB and MG) the abbreviations should be explained.
Comment 2
Line 76 What was conductivity of deionized water?
Comment 3
Line 82 What metal was used in the metal wire?
Comment 4
Line 137 “Shimadza” it is mistake. It should be Shimadzu.
Comment 5
Line 151 “Shimadza” it is mistake. It should be Shimadzu.
Comment 6
Line 242 What causes a reduction of Ti(IV) to Ti(III)?
Comment 7
Line 323 At what wavelength were the different dyes determined (MB, MO, RhB, MG)?
Comment 8
Are dye (MB, MO, RhB, MG) degradation products safe for living organisms and plants?
Comment 9
What are the degradation products of dye degradation (MB, MO, RhB, MG)?
Comment 10
Line 148. The initial pH value was 7. What was the pH value after dye degradation (MB, MO, RhB, MG)?
Author Response
Thank you for your careful review comments, which made the article even better. I apologize for taking up a lot of your time with too many faults in the manuscript.
Point 1: Line 17 (MB, MO, RhB and MG) the abbreviations should be explained.
Response 1: The abbreviation has been changed to the full name to avoid misunderstandings. A screenshot of the revised manuscript is as follows.
Point 2: Line 76 What was conductivity of deionized water?
Response 2: The deionized water used for the experiments was purchased commercially without further treatment. Unfortunately, the laboratory does not have equipment to measure conductivity, so the conductivity of deionized water was not measured.
Point 3: Line 82 What metal was used in the metal wire?
Response 3: The metal wire is copper, which is relatively chemically stable and easily available.
Point 4: Line 137 “Shimadza” it is mistake. It should be Shimadzu.
Response 4: Thank you for your careful review. The error has been corrected. A screenshot of the revised manuscript is as follows.
Point 5: Line 151 “Shimadza” it is mistake. It should be Shimadzu.
Response 5: T Thank you for your careful review. The error has been corrected.
Point 6: Line 242 What causes a reduction of Ti(IV) to Ti(III)?
Response 1: There is actually no significant reduction of Ti(IV) to Ti(III), only a small amount. We did not do the relevant tests, so we did not analyze the alignment reaction mechanism and Ti(III) ratio. A relevant reference was found to have an explanation for the reduction of Ti(IV) to Ti(III). This issue has not been studied in depth because there is no more experimental data to support it. A screenshot of the found literature is shown below.
Point 7: Line 323 At what wavelength were the different dyes determined (MB, MO, RhB, MG)?
Response 7: The maximum absorption peaks of MB, MO, RhB, and MG were 665 nm, 466 nm, 553 nm, and 616 nm, respectively.
Point 8: Are dye (MB, MO, RhB, MG) degradation products safe for living organisms and plants?
Comment 9
Response 1: We believe that the degradation products should be considered safe. Many studies reported the use of TiO2 photodegradation of MB, MO, RhB, MG to treat water pollution problems, so the products were not analyzed for biosafety. Unfortunately, our laboratory is also not equipped for biosafety research and we are unable to go into further research on biosafety.
Point 9: What are the degradation products of dye degradation (MB, MO, RhB, MG)?
Response 1: I am very sorry that the experiments were not analyzed for degradation products and no experimental data are available.
Point 10: Line 148. The initial pH value was 7. What was the pH value after dye degradation (MB, MO, RhB, MG)?
Response 1: Please forgive us for removing the expression that the PH value is 7. This is a mistake in writing. Thank you very much for asking the question and giving us the opportunity to discover this almost inexcusable error. Since acetic acid was used during the hydrothermal process, the PH value was tested for concern about acetic acid residue during washing with water. PH of 7 is the wash water of the catalyst. The pH was not specifically adjusted for the simulated contaminants in the experiment. No testing was done either. As a writer, I sincerely apologize.
Reviewer 5 Report
The manuscript "Synergistic effects of multiple heterojunctions and dopant atom enhance the photocatalytic activity of C modified Zn-doped TiO2 ternary composite nanofiber film" is quite interesting. However, this manuscript needs some improvement to be published in the Coatings journal. Here are some improvements that need to be considered:
1- The manuscript shows around 27% plagiarism. Kindly see the similarity report.
2- The abstract should be modified to give a clear idea of the results.
3- The author should represent and highlight the novelty of the work in the manuscript.
4- The authors didn't mention the equations used in calculating the efficiency in the photocatalytic part in the experimental section. Check the following references that help in this:-
-Fast and Excellent Enhanced Photocatalytic Degradation of Methylene Blue Using Silver-Doped Zinc Oxide Submicron Structures under Blue Laser Irradiation.
- Facile synthesis of Y2O3/CuO nanocomposites for photodegradation of dyes/mixed dyes under UV- and visible light irradiation.
5- The authors didn't mention the equations used in calculating the structural parameters like crystallite size, lattice strain, cell volume, and dislocation density for the prepared samples in the XRD section. Check the following references that help in this:-
-Enhancement in the Structural, Electrical, Optical, and Photocatalytic Properties of La2O3-Doped ZnO Nanostructures.
- Electrocatalytic Degradation of Rhodamine B Using Li-Doped ZnO Nanoparticles: Novel Approach
6- The authors didn't mention the equations used in calculating the optical band gap for the prepared samples in the UV-vis DRS. Check the following references that help in this:-
- Impact of Mo-Doping on the Structural, Optical, and Electrocatalytic Degradation of ZnO Nanoparticles: Novel Approach.
- One-pot synthesis of multifunctionalized Nd2O3 dispersed ZnO nanocomposites for enhancing electrical, optical, and photocatalytic applications
7- The quality of the images is very low; kindly change the images to high-resolution ones.
Author Response
Thank you for your careful review comments, which made the article even better. I apologize for taking up a lot of your time with too many faults in the manuscript.
Point 1: The manuscript shows around 27% plagiarism. Kindly see the similarity report.
Response 1: Thanks for your reviewer. We rewrote some of the content to reduce the repetition rate. The turnitin database was used for the duplication rate detection and the similarity was 13%. The similarity will vary due to the different databases used. Avoiding repetition is difficult in many places, such as instrumentation information, peak locations, fixed chemical bonds, 发ormula description and some proprietary nomenclature. These contents have a relatively high percentage of similarity. This information is useful for the reader to understand the materials, apparatus, and methods used in the experiments and should not be removed. A number of unchangeable areas were screenshotted, and I hope these do not affect the publication of the manuscript.
Point 2: The abstract should be modified to give a clear idea of the results.
Response 2: To to give a clear idea of the results, we modified the abstract and added a result oriented description. A screenshot of the revised manuscript is as follows.
Point 3: The author should represent and highlight the novelty of the work in the manuscript.
Response 3: We revised the abstract, introduction and mechanism of the manuscript in the hope of better highlighting the novelty of the work and the improvements with respect to the state of the art. Screenshots of the revised manuscript is as follows.
Abstract
Introduction
mechanism
Point 4: The authors didn't mention the equations used in calculating the efficiency in the photocatalytic part in the experimental section. Check the following references that help in this:
-Fast and Excellent Enhanced Photocatalytic Degradation of Methylene Blue Using Silver-Doped Zinc Oxide Submicron Structures under Blue Laser Irradiation.
- Facile synthesis of Y2O3/CuO nanocomposites for photodegradation of dyes/mixed dyes under UV- and visible light irradiation.
Response 4: Thank you for recommending a quality reference, we have added the formula and cited this reference. A screenshot of the revised manuscript is as follows.
Point 5: The authors didn't mention the equations used in calculating the structural parameters like crystallite size, lattice strain, cell volume, and dislocation density for the prepared samples in the XRD section. Check the following references that help in this:-
-Enhancement in the Structural, Electrical, Optical, and Photocatalytic Properties of La2O3-Doped ZnO Nanostructures.
- Electrocatalytic Degradation of Rhodamine B Using Li-Doped ZnO Nanoparticles: Novel Approach
Response 5: We calculated the crystallite size, lattice strain, and dislocation density and cited the recommended quality references. A screenshot of the revised manuscript is as follows.
Point 6: The authors didn't mention the equations used in calculating the optical band gap for the prepared samples in the UV-vis DRS. Check the following references that help in this:-
- Impact of Mo-Doping on the Structural, Optical, and Electrocatalytic Degradation of ZnO Nanoparticles: Novel Approach.
- One-pot synthesis of multifunctionalized Nd2O3 dispersed ZnO nanocomposites for enhancing electrical, optical, and photocatalytic applications
Response 6: Thank you for the quality references, which has been cited in the manuscript.
Point 7: The quality of the images is very low; kindly change the images to high-resolution ones.
Response 7: It is very unfortunate that the SEM we received was not of high resolution. The samples were not recovered after testing, so they could not be tested again. We added small magnifications to SEM photos, while the photo brightness and contrast were adjusted in the hope of improving the clarity of the photos. It is expected that these measures will improve the reader's experience and help the reader to have a more comprehensive impression of material morphology.
A screenshot of the revised manuscript is as follows.
Round 2
Reviewer 3 Report
In the revised version of the submitted manuscript the authors faced most of the lacks underlined in the first round and significantly improved the paper. Nevertheless, several important remarks were deal without significant attention. The authors were very honest in their replies and this is a good quality, but they declare some backgrounds that make unacceptable the paper. In the following points (not in priority order) I reply to the authors answers to my observations.
1) The paper title appears more suitable in the new version.
5) The cooling time is not a problem. How long the samples were kept at 500°C?
9) Authors could specify that the employed amount (1g/L) was the best compromise for an optimal kinetic study. Concerning pH (“pH”, not “PH”), authors could specify that the pH was the natural pH of the dyes solution, without any further modification.
The authors comment about this point cannot be accepted at all. Authors substantially declare that no temperature controls were available during the photocatalytic tests and this is unacceptable for a good data explanation and experiment replication. Starting from the authors statements, it is impossible do not suppose that the temperature changed also during experiments, taking into account that a 350W Xenon lamp was lighting the reactor. Authors need a thermostatic bath for their incoming experiments.
10) The Pt coating was not mentioned in the SEM experiment in the first version, neither in this new one. The answers for the referees’ comments are not for the referees only, but help all the readers.
I cannot accept the unavailability of the samples for further analyses. No scientists eliminate the samples before the data publication.
11) Which error bars?
Concerning the Scherrer’ dimensions, density and strain, these parameters are given with 8-10 decimal place: this is not physically acceptable.
14) XPS analyses. I understand the authors reasons about the cost of the analyses, but I cannot accept this answer for justifying a data lack in a paper. The quality of the paper is affected by this and no complete reasoning can be done (i.e.: authors cannot compare the carbon signals in C-containing and not C-containing samples, and this element is crucial for the invoked mechanism). Authors themselves wrote about adventitious carbon, so it appears licit asking for the quantification among adventitious and not adventitious carbon. The limited time of the revision cannot be invoked as a justification: an extra-time can be asked; the paper quality comes first.
Concerning the hydroxyls presence after thermal treatment, their presence should not be surprising. Hydroxyls presence is detected also in samples treated at higher temperature.
20) Error bars have not been added in the histograms; the values are simply written beside the sample labels. Moreover, the declared errors appear very low: how are they determined?
21) The comment is only partially answered. The spectra of the dye solutions during the experiments should be already available, I hope. Concerning the filters for shielding the lamp in UV and visible spectra I agree that this point is not easily overcame, but the manuscript should then be integrated with dedicated considerations about this. The samples do not absorb significantly in the visible range, so the activation is borderline among UV and visible edge, so without a clear indication about the activation requirements.
22) No new experiments are need in this case. Authors just need to calculate the kinetic constants from the experiments already done. Comparing the kinetic constant, in my opinion, is more useful then the trend comparison, better highlighting possible variation in their values (in the error confidence, of course).
23) In their reply, authors state that the mass of the recovered samples was not weighted. So, how can they perform the next cycle? How can they dose the catalyst?
Author Response
Thank you for taking so much time to help improve the quality of the manuscript. You even help the author write the sentences that should be used in the manuscript. Your patience and seriousness are admirable. Your knowledge of experimental design, data processing and description of experimental phenomena is impressive, and your suggestions for improving the manuscript would be very helpful. We have done our best to revise the manuscript and hope that it will meet the publication standards.
Point 1: The paper title appears more suitable in the new version.
Response 1: The title was improved thanks to your suggestion, otherwise it would have become a big problem after publication. I would like to express my sincere gratitude for your kind guidance.
Point 5: The cooling time is not a problem. How long the samples were kept at 500°C?
Response 5: Please forgive me for misinterpreting your review, which may be due to the different language habits of the countries. The samples were held at 500°C for 1h. A screenshot of the revised manuscript is as follows.
Point 9: Authors could specify that the employed amount (1g/L) was the best compromise for an optimal kinetic study. Concerning pH (“pH”, not “PH”), authors could specify that the pH was the natural pH of the dyes solution, without any further modification.
The authors comment about this point cannot be accepted at all. Authors substantially declare that no temperature controls were available during the photocatalytic tests and this is unacceptable for a good data explanation and experiment replication. Starting from the authors statements, it is impossible do not suppose that the temperature changed also during experiments, taking into account that a 350W Xenon lamp was lighting the reactor. Authors need a thermostatic bath for their incoming experiments.
Response 9: Thank you for teaching me to write in a standardized language and helping me perfect my paper. I think I used the wrong expression to make you misunderstand in the temperature control. We had no way to control the experimental temperature at 20-25 °C, in the absence of air conditioner. No other cooling equipment available. The indoor temperature was higher than 25°C in summer. There was no way to lower the temperature for the experiment, so the temperature was kept at 30 °C above room temperature. Thermostatic bath was used in experiments. However, the thermostatic bath can only raise the temperature, not lower it. This part of the description was added. A screenshot of the revised manuscript is as follows. "We don't have better experimental conditions." means that unfortunately there is no way to cool down the experiment. I apologize for the misunderstanding caused by the unclear description.
Point 10: The Pt coating was not mentioned in the SEM experiment in the first version, neither in this new one. The answers for the referees’ comments are not for the referees only, but help all the readers.
I cannot accept the unavailability of the samples for further analyses. No scientists eliminate the samples before the data publication.
Response 10: I apologize for neglecting to revise in the manuscript due to my carelessness. We didn't coat the samples with metal in the experiment. Samples are mailed to a commercial testing company. Inorganic materials usually cannot be directly tested by SEM due to their poor electrical conductivity. The samples were coated with Pt by tester from commercial company. We have added the necessary descriptions. A screenshot of the revised manuscript is as follows. We did not discard the samples intentionally. The sample was destroyed in an accident. All data are from before the accident. We did not ask a commercial testing company to help us keep the samples because keeping of samples requires additional fees and accidents are unpredictable. In fact, for some irresistible reasons, I will not be able to work in the lab for a long time. I'm sorry to give you this response. That's sad to hear.
Point 11: Which error bars?
Concerning the Scherrer’ dimensions, density and strain, these parameters are given with 8-10 decimal place: this is not physically acceptable.
Response 11: Error bars are added. The number of decimal places after the decimal point of the parameters are changed according to the physics error calculation rule. The physics requires that the number of decimal places of the parameter be the same as the first significant digit of the error. r has more decimal places, because the error is too small.
Point 14: XPS analyses. I understand the authors reasons about the cost of the analyses, but I cannot accept this answer for justifying a data lack in a paper. The quality of the paper is affected by this and no complete reasoning can be done (i.e.: authors cannot compare the carbon signals in C-containing and not C-containing samples, and this element is crucial for the invoked mechanism). Authors themselves wrote about adventitious carbon, so it appears licit asking for the quantification among adventitious and not adventitious carbon. The limited time of the revision cannot be invoked as a justification: an extra-time can be asked; the paper quality comes first.
Concerning the hydroxyls presence after thermal treatment, their presence should not be surprising. Hydroxyls presence is detected also in samples treated at higher temperature.
Response 14: A method to calculate the ratio of useful C to useless C was figured out, which is exciting and exhilarating. We can solve this problem based on the available data. Otherwise a lot of work will become futile. The atomic percentage of C in the EDS was calculated as the percentage of "useful" C. The sample is covered by Pt during the test, which reduces the contamination of air C. In fact, this is only an estimate, since it is difficult to guarantee that the C contamination is the same for different samples. The same sample will also have different C percentages depending on the time of exposure to air in XPS test. I think you're right. Such an estimate is necessary. Clarify the source and approximate percentage of C in XPS so that the reader understands the role of the split peaks. This is what must be done to interpret the data. Thank you for taking so much time to help us correct the manuscript, even for spelling mistakes. It makes the paper show our work better.
Until I get a positive answer from you, I'm really not sure about the presence of hydroxyl at high temperatures. This has given me a deeper understanding of the composition of the material. The hydroxyl groups have been added in XPS. Screenshots of the revised manuscript is as follows.
Point 20: Error bars have not been added in the histograms; the values are simply written beside the sample labels. Moreover, the declared errors appear very low: how are they determined?
Response 20: By fitting the line directly in origin software, the standard error is calculated directly, without additional calculations. There seems to be no arithmetic process that can be written into the manuscript. I guess that you want the role of these error bars to be described. I have added some of these descriptions. A screenshot of the revised manuscript is as follows. In fact I never noticed these errors. Almost no comments on errors can be found in the papers on photocatalytic calculation of k values. There is no similar view to help me understand these errors or give relevant comments. I have repeatedly studied the values of standard errors. They seem to be independent of the proximity of the line and points. It seems that errors are more likely to be low for small values of k. The error formed by the straight-line fit seems to be calculated differently from the general error. I searched for some mathematical principles. Hopefully, the math helped me understand the meaning of these errors, but I failed. Origin uses a mathematical method known as the Generalized Least Squares Method to perform very complex operations. I'm guessing this is how origin fits a straight line, but not sure. I am unable to understand these complex mathematical principles. So only a few notes can be added. Screenshots of the revised manuscript is as follows.
Point 21: The comment is only partially answered. The spectra of the dye solutions during the experiments should be already available, I hope. Concerning the filters for shielding the lamp in UV and visible spectra I agree that this point is not easily overcame, but the manuscript should then be integrated with dedicated considerations about this. The samples do not absorb significantly in the visible range, so the activation is borderline among UV and visible edge, so without a clear indication about the activation requirements.
Response 21: Maybe I didn't understand your comment too well. To determine the maximum absorption wavelength of the four dyes, the UV-visible spectra of four dyes were measured. The maximum absorption peaks of MB, MO, RhB, and MG were 665 nm, 466 nm, 553 nm, and 616 nm, respectively. No degradation of the dye was observed when light or photocatalyst was absent. The shorter adsorption time under darkness is due to the fact that no significant degradation was found in the pre-experiments, with adsorption over 30 min. No self-decomposition of dyes was observed. I guess the dye does not fade easily, otherwise it cannot be used for dyeing. Some of these descriptions were added to the manuscript. A screenshots of the revised manuscript is as follows.
Regarding the role of UV-visible light:
Please allow me to use your sentence directly, as it is very scientific and accurate. I could not have written it better. I believe this sentence was written especially for my manuscript to help refine my writing.
Point 22: No new experiments are need in this case. Authors just need to calculate the kinetic constants from the experiments already done. Comparing the kinetic constant, in my opinion, is more useful then the trend comparison, better highlighting possible variation in their values (in the error confidence, of course).
Response 22: Thank you for your pertinent advice! Figure with error bars have been added to the manuscript.
Point 23: In their reply, authors state that the mass of the recovered samples was not weighted. So, how can they perform the next cycle? How can they dose the catalyst?
Response 23: The recovered photocatalysts are washed and dried and used directly for the next cycle of experiments. The same volume of organic dye was used for each cycling experiment. We initially thought the photocatalyst is not easily lost in cycling experiments due to the shape of a small film, not a powder. With your prompting, we have realized that this is not strict enough. In future experiments, we must pay attention to the rigor of each step of the experiment. Consider all possible scenarios thoroughly. Despite such shortcomings in the experiment, we hope that the preceding work will be accepted. A description of the cyclic experimental procedure has been added to the manuscript. A screenshots of the revised manuscript is as follows.
Reviewer 5 Report
The authors have revised the manuscript according to the comments. I feel that the paper can be published in its present form. Thanks

Author Response
Thank you for posting the shape-similarity report. It's really hard to reduce the duplication rate. Your suggestions helped a lot with the manuscript. I really appreciate it.